# Individual, family, school and neighborhood predictors related to different levels of physical activity in adolescents: A cross-sectional study

**Isabella Toledo Caetano**[1◉], **Fernanda Karina dos Santos**[1◉], **Alynne Christian Ribeiro Andaki**[2‡], **Thayse Natacha Q. F. Gomes**[3,4◉*], **Paulo Roberto dos Santos Amorim**[1‡]

**1** Department of Physical Education, Federal University of Viçosa, Viçosa, Minas Gerais, Brazil,
**2** Department of Sports Science, Postgraduate Program in Physical Education, Federal University of Triangulo Mineiro - UFTM, Uberaba, Minas Gerais, Brazil, **3** Department of Physical Education and Sports Science, Health Research Institute, Physical Activity for Health Research Cluster, University of Limerick, Limerick, Ireland, **4** Federal University of Sergipe, São Cristóvão, Sergipe, Brazil

◉ These authors contributed equally to this work.
‡ ACRA and PRSA also contributed equally to this work.
* thayse_natacha@hotmail.com

**Data Availability Statement:** The minimum set of underlying data of our study called "Supporting information Files" was uploaded. The data will be

## Abstract

The aim of this study was to investigate the association among individual, family, school environment and neighborhood predictors with the different levels of physical activity (PA) [light (LPA) and moderate to vigorous PA (MVPA)] in Brazilian adolescents. A cross-sectional study was carried out with 309 adolescents with a mean age of 15.37 (± 0.57) years. PA and sleep time were assessed by accelerometry. Individual predictors were determined by anthropometry and questionnaires, while family, school environment and neighborhood predictors were assessed using questionnaires. Robust Regression analysis was performed considering a significance level of 5%. Individual and environmental variables were able to respectively predict 64% and 13.6% of adolescents' participation in LPA. Work ($\beta_p = 0.2322$), gender ($\beta_p = -0.1318$), commuting to school ($\beta_p = -0.1501$), sleep ($\beta_p = -0.1260$) and paved roads ($\beta_p = -0.1360$) were associated with LPA. It was also observed that individual (59.4%) and environmental (27.4%) variables were able to predict adolescents' participation in MVPA. Work ($\beta p = 0.1656$), commuting to school ($\beta p = 0.1242$) and crime ($\beta p = 0.1376$, and gender ($\beta p = -0.3041$) and paved roads ($\beta p = -0.1357$ were associated with MVPA. Such results indicated that boys, those who work and those who live in unpaved neighborhoods presented greater time in LPA and MVPA; those who live in neighborhoods with higher crime had higher time spent in MVPA; and those who passively commute to school had more time in LPA. There was an average reduction of 5.0 minutes in LPA time for each additional hour of sleep. Finally, students who actively commute to school had more time in MVPA. Individual factors and those related to the neighborhood environment can play an important role in understanding the variables which can influence the different levels of PA in adolescents.

available in the "Public Repository" (https://data.
mendeley.com/datasets/gxcs8wzcfb/4).

**Funding:** This study was financed in part by the
Coordination for the Improvement of Higher
Education Personnel – Brazil (CAPES) – PNPD –
CAPES. The funders had no role in study design,
data collection and analysis, decision to publish, or
preparation of the manuscript.

**Competing interests:** The authors have declared
that no competing interests exist.

## Introduction

It is widely recognized that physical activity (PA) positively affects the health of adolescents
[1, 2], and consistent evidence has been reported regarding the benefits of moderate to vigorous intensity PA (MVPA) [3, 4]. However, recent evidence has highlighted the health benefits
of light PA (LPA), defined as activities with an energy expenditure between 1.5–3.0 METs,
especially given its proximity to daily activities, such as walking [5, 6]. Although there are no
specific recommendations for LPA [7], previous studies with adolescents found a positive association between LPA and cardiometabolic markers [5, 7, 8], as reported by Ayala et al. [8]
when evaluated 219 adolescents, in which a moderating effect of LPA time on seated activity
time was found; in addition, adolescents who performed more than 300 minutes of LPA daily
had lower values of adiposity markers (body mass index and waist circumference).

The health benefits of MVPA have been widely reported, however a decline in PA levels has
been observed in the pediatric population [9–11]. Global estimates show that more than three-quarters (81%) of adolescents do not meet global PA recommendations [2]. According to the
World Health Organization (WHO), adolescents should practice, on average, at least 60 minutes of MVPA per day and incorporate at least 3 days per week of vigorous-intensity aerobic
activities, as well as those that strengthen muscles and bones [10]. It is worth noting that PA is
a complex behavior, influenced by multiple factors [12–14], and understanding the predictors
that can influence the involvement of PA in adolescents can help develop strategies which promote increases in the different PA levels.

In this context, the social ecological model presents itself as a relevant theoretical basis for
understanding the complexity of factors associated with PA [15], as it suggests reciprocal interactions between the subject and various levels of influence, including intrapersonal (biological,
behavioral, psychological), interpersonal (social and cultural), environmental (social and physical/infrastructural—built and natural), organizational and political domains [12, 15].

In this sense, previous studies have reported that individual variables such as gender and
age [13, 16, 17], sleep time and psychological/cognitive/emotional factors [12, 18], and high
body mass index (BMI) [18] are presented as predictors of PA in adolescents. In addition, families are important PA providers, especially parents [19], including actions such as family support and company for PA [20], parental education and occupational history [12, 13, 18, 21],
and socioeconomic status [16, 22]. In addition to the family environment, the school environment is a place where adolescents spend a considerable part of the day and offers diversified
opportunities for PA, such as active transport to and from school, physical education classes,
sports activities, sports equipment and recreation [12, 23, 24]. The neighborhood also seems to
play an important role in the involvement with PA, as studies indicate that the proximity of
home to green spaces [25], and greater ease of walking in the neighborhood [26], street lighting, paved streets and cycling paths [27] are all associated with the involvement in MVPA
practiced by adolescents.

Multiple predictors have been consistently reported in the literature that can influence
MVPA, active transport-related PA or leisure-time PA in general in the pediatric population
[12, 21, 28]. However, in the literature it is still unclear which predictors can influence participation in the LPA. Thus, it is important to investigate a set of environmental predictors, meaning the different contexts that the individuals are inserted, such as their homes, schools and
neighborhoods in the LPA. Given the complexity of the PA phenotype, our hypothesis is that
each PA level can be influenced by a combination of different predictors. Thus, the present
study aimed to investigate the association among individual, family, school environment and
neighborhood predictors with the LPA and MVPA in adolescents from the city of Viçosa, MG,
Brazil.

## Methods

### Study design and participants

This is a cross-sectional study with a random sample of adolescents of both genders aged between 14 and 16 years, regularly enrolled in the 1st year of high school in public schools (state and federal) in the city of Viçosa, Minas Gerais, Brazil. Data were collected between March and December 2019.

Public schools which offered high school were consulted to inform the number of students enrolled in the studied age group. There were 968 students enrolled in 7 schools in 2019. Based on this information, the sample size was calculated using the Stat-Calc of the EpiInfo version 7.2.2.16 software program (Georgia, United States). The sample size calculation considered a confidence level of 95%, prevalence of 50% [29, 30], an acceptable error of 5% and design effect of 1.1. The result presented for the minimum sample size was 305 individuals. Next, 20% was added to this calculation in case of possible losses, totaling a minimum sample of 366 adolescents. Then, 6 of the 7 public schools (5 states and 1 federal) were selected to participate in the study. Students from each school were selected by a draw based on the list of students enrolled [31].

The students had to present an Informed Consent Form (ICF) and an Informed Assent Term (IAT) duly signed by their legal guardians and by the adolescents to participate in the study. Exclusion criteria were: pregnancy, participation in a weight reduction or weight control program, physical disability that limited PA participation, temporary or permanent mental disability, regular use of diuretics/laxatives, and not having valid accelerometer information.

The study was conducted in accordance with the guidelines of the Declaration of Helsinki and all procedures were approved by the Ethics Committee in Research involving human beings.

### Physical activity and sleep

ActiGraph GT3X accelerometer was used to monitor the time spent in light PA (LPA) and moderate-to-vigorous PA (MVPA) (min.day$^{-1}$), and the sleep/wake time (hours.day$^{-1}$). The ActiLife software program (version 6.13.4) (ActiGraph, LLC, Fort Walton Beach, USA) was used to perform all analyses. The adolescents wore the monitors attached to an elastic belt over their right hip for 8 consecutive days, including during nighttime sleep. They were instructed not to change their daily routine and the accelerometer was to be removed only for water activities. Each subject received a diary for the equipment use in which they should write down the time they woke up and what time they went to sleep at night every day, in addition to the times when the monitor was removed and then put back on the body. The first day of use (the day the equipment was received) was not considered in the analysis to avoid the Hawthorne Effect [32].

The accelerometer was initialized to collect data at a sampling rate of 30 Hz with normal filter in 1s epochs and then the data was reintegrated into 15s epochs. Non-usage time was defined as zero consecutive counts/minute lasting at least 20 minutes. The participants had to reach a minimum of 10 h.day$^{-1}$ of "usage time" [33] on at least 5 days of the week, of which at least 1 day should be a weekend day in order to be included in the analysis. Sleep/wake time analyzes were performed from accelerometer information, such as daily graphs and inclinometer data, together with the aid of the accelerometer usage diary (time you went to sleep/time you woke up) completed by the adolescents. The cut-off points developed by Romanzini et al. [34] validated for Brazilian adolescents, using a magnitude vector and 15s epochs were used to classify PA.

The continuous values of the time of LPA and MVPA in minutes considered for the analysis.

## Independent variables

A total of 23 possible factors associated with PA were selected based on reviews [12, 21, 28]. Thus, the variables were structured into influence blocks, meaning that the variables which could influence in the LPA and MVPA by adolescents were grouped.

First, intrapersonal relationships (the subject and their individual predictors) were considered. Then, interpersonal relationships were determined, which included the interaction between the subject and their family, followed by the interaction between the subject and the school environment. Finally, the outermost layer, the relationship between the subject and the built environment of the neighborhood [15, 35, 36]. The four blocks were: (1) Individual predictors (demographic, biological, psychological and behavioral); (2) Family predictors (head of the household health behaviors and socioeconomic status); (3) School environment predictors (school structures for PA, incentives for PA practice); and (4) Neighborhood predictors (physical structures of the neighborhood and places intended for PA practice).

It is important to emphasize that the first category for all the variables that will be described below was considered the reference category in the analyzes and was determined as the one which presents the most unfavorable values for each variable or the absence of a certain behavior/characteristic, except for the gender variable.

## Individual predictors

Body fat (BF) was evaluated through triceps and medial calf skinfold measurements obtained from the right side of the body, according to the guidelines established by the International Society for the Advancement of Kinanthropometry [37], making sure that the participant had not performed any type of exercise for a minimum period of 4 hours before the assessment. Each measurement was taken three times alternately, and the measurement was repeated in case of a difference greater than 5% between the three values. The average of the two closest values was used for the analyses. The body fat percentage (BF%) was estimated by the equation of Slaughter et al. [38] and classified according to Lohman et al. [39] as underweight, eutrophic, risk of overweight and overweight. The categories were dichotomized (risk of overweight/overweight and underweight/eutrophic) for the analyses.

**Biological predictors, life habits and psychological predictors.** Biological predictors such as age and gender, lifestyle habits related to alcohol consumption, work, commuting to school and sitting time, as well as psychological predictors, such as stress level and feeling of loneliness, were evaluated by questions specific to the questionnaire "Risk Behaviors of Adolescents" (*COMPAC*) [40].

Students should report their age in years, and gender (male or female). Alcohol consumption was evaluated by the question: "During a normal (typical) week, do you consume alcoholic beverages?", and the answers were recategorized as: 'does not consume alcohol' and 'consumes alcohol' for those who responded to consume at least one dose/week or more. The adolescents also answered the following question in relation to work: "Do you work?" The answers were recategorized into: 'I don't work'; and 'yes, I work' for adolescents who work 20 hours/week or more. The way of commuting to school was evaluated by the question: "How do you normally commute to go to school?"; the answers were recategorized into: 'passive commuting' for students who used a motorcycle, car or bus; and 'active commuting' for those who commuted on foot or by bicycle. Total sitting time was assessed by the question: "How much time do you spend sitting, talking to friends, playing cards or dominoes (or other

games), talking on the phone, in traffic as a passenger, reading or studying?". The weighted average of hours sitting on weekdays and weekends enabled estimating sitting time on seven days of the week and were used in the analyses.

Next, the students answered the following question for the feeling of loneliness: "During the last 12 months, how often did you feel lonely?"; the answers were recategorized into: 'always lonely', 'sometimes lonely' and 'never lonely'. The level of stress was assessed by the question: "How would you describe the level of stress in your life?"; the responses were recategorized into: 'always stressed', 'sometimes stressed' and 'no stress'.

Total screen time (ST) was assessed by the Portable Technologies and Mobile Internet Questionnaire [41]. Participants were asked: "How much time per day do you spend on average accessing the internet through [technology of interest]?", answered for a normal weekday and a weekend day. A score in minutes was created for each type of portable technology (cell phone, tablet, and laptop) evaluated. The individual scores were created based on the time-weighted average of the technology of interest considering the five weekdays and two weekend days. The scores in minutes of the three technologies were added to verify the total ST and used in the analyses. The time in hours of daily ST was considered for the analyses.

Finally, information regarding sleep time was measured by an ActiGraph GT3X accelerometer as described above. The average time in hours of sleep of all the days evaluated were considered for the analyses.

## Family predictors

Family predictors were obtained through questionnaires completed by the legal guardian, called the "head of the household" who signed the ICT.

The socioeconomic status (SS) of the family was extracted from the Economic Classification Criteria of the Brazilian Association of Research Companies [42]. According to the final score, the participants' SS was classified into 3 classes: 'high' (classes A and B1), 'average' (B2 and C1) and 'low' (C2 and D-E).

The household head PA was evaluated by the short version of the International Physical Activity Questionnaire (IPAQ) [43]. Only information referring to MVPA was used for the present study, in which the internationally accepted cut-off point of at least 150 minutes of MVPA per week was adopted [2]. Then, they were categorized into: 'insufficiently active' and 'active'.

For the BMI assessment, the household head should self-report body mass and height while completing the Diabetes Risk Score—Findrisk questionnaire [44]. The BMI calculation of the household head was performed by the evaluators. Then, those with BMI values $\geq 25$ Kg/m$^2$ were classified as 'high BMI', and those with BMI values $< 25$ Kg/m$^2$ were classified as 'adequate BMI' [45].

## School environment predictors

The school environment predictors were obtained by a session of the COMPAC questionnaire [40], filled in by the students. The students answered the following question regarding the number of physical spaces for physical education (PE): "How many physical spaces for sports and PE are there in your school?"; the answers were then organized as: 'only 1 space'; and '2 or more spaces'. To the offer of extracurricular PA was asked: "Does your school offer sports activities (not including PE classes)?", whose answers were: 'no' or 'yes'. Students were also asked about "How many PE classes do you participate in during a normal (typical) week?"; the response options were: 'I don't participate', 'up to 2 classes/week' and '2 or more classes/week'.

## Neighborhood predictors

The neighborhood environment was assessed using the Neighborhood Walkability for Youth (NEWS-Y) scale [46], adapted for Brazilian adolescents [47]. The NEWS-Y assess adolescents' perception of the predictors of the neighborhood environment which may be associated with walking and other types of PA [48, 49]. The questions are related to the place to walk, ease of walking to the bus stop, paved roads, street lighting, crime and travel time to the bus stop were used in the present study. An average was calculated for each of the domains, so that values greater than the average indicated higher values in the respective domain [46, 50]. Then, the environment predictors—place to walk, ease of walking to the bus stop, paved roads, street lighting, crime—were categorized as 'no', referring to the absence of such attribute in the neighborhood environment, and 'yes', for the presence of the attribute. The answer options for the travel time to the bus stop were recategorized into: 'long', which corresponds to the travel time between 11 and 30 minutes together with the answers "I don't know/doesn't exist"; 'medium' for the time between 6 to 10 minutes; and 'short' for the time between 1 to 5 minutes.

## Data collection procedures

The evaluations were conducted by a previously trained team. Data collection took place in four meetings with each participant. The adolescents received information related to the research and its procedures, were invited to participate and received the ICF and IAT in the first meeting. In the second meeting, the signed terms were given to the researchers and the students received verbal guidance about the questionnaires before filling them out. The questionnaires were individually completed in the classroom with the help of the first author of the study, in case of doubts. In the third meeting, skinfolds were measured and the accelerometer was placed for direct assessment of PA and sleep time, being held in a private room provided by the schools. Each student received a sheet with equivalent instructions on the use of the device and an accelerometer use diary, which should be completed. In this same meeting, each student was given the questionnaires to take home which should be filled in by the parents/legal guardians. In the fourth meeting, the students should return with the completed questionnaires and hand in the accelerometer along with the usage diary.

The minimum set of underlying data of our study called "S1 File" was uploaded. The data will be available in the "Public Repository".

## Statistical analysis

Statistical analyzes were performed by the Statistical Package for the Social Sciences (SPSS), version 20.0 (IBM Corporation®, New York, USA) and Software for Statistical and Date Science (STATA) version 13.0 (StataCorp LP®, Texas, USA).

To evaluate the normality, was used the Kolmogorov-Smirnov that showed absence of normality in the distribution of continuous variables. Therefore, the results were presented based on the median and interquartile range (IQ). The Mann-Whitney test was used to verify the difference between the genders of the variables related to the LPA and MVPA. Bonferroni adjustment was used in the two-by-two post hoc tests to verify the difference between the groups.

Spearman's correlation coefficient was used to verify the degree of statistical dependence between the analyzed variables (the correlation coefficient values can be found in S1 Table). The simple linear regression model was used to verify how much each variable contributed to the LPA and MVPA. The variables which obtained a significance value of $p < 0.1$ were included in the subsequent analyses of Robust Linear Regression [51, 52] to verify the association with the PA levels. The variables were inserted into prediction blocks (models).

Statistical modeling was based on a block of predictors approach: Block 1—individual variables (biological predictors, life habits and psychological predictors); Block 2—Block 1 and family variables; Block 3—Block 2 and environment variables; Block 4—Block 3 and neighborhood variables. Thus, the change significance in the F statistic of the analysis of variance of each model was evaluated to verify which model best fits the variables added in each block ($p \leq 0.05$). A new analysis was performed after model 4, considering only the variables that were significant in model 4. Thus, all variables that had statistical significance ($p \leq 0.05$) were included in the final model referring to the LPA and MVPA. The significance of the final model was evaluated by the F-test and the goodness of fit by the coefficient of determination ($R^2$).

## Results

A total of 367 adolescents participated in the study, but 58 were excluded from the analysis process because they did not have valid data regarding the accelerometer. Therefore, the final sample consisted of 309 adolescents, of which 57% were female, with a mean age of 15.37 (± 0.57) years.

Table 1 presents the sample predictors, with the results of the continuous variables and categorical variables analyzed. There were significant differences between sex for LPA and MVPA, with higher values observed in the males group.

Significant differences between the sex were observed for individual variables (work, commuting to school, BF, stress, loneliness), family variables (household head MVPA), school environment (frequency of PE classes) and neighborhood (lighting). A greater proportion of girls: do not work, do not actively go to school, are at risk of being overweight/overweight, reported feeling lonely at times, are from families whose household head is physically inactive, participate in up to 2 PE classes/week and live in well-lit neighborhoods. In relation to boys, a higher proportion reported being always stressed and participating in more than 2 PE classes/week.

Table 2 shows the simple associations among potential variables (individual, family, school environment and neighborhood) and each outcome variable (LPA and MVPA). Positive associations were found between LPA and work, alcohol, frequency of PE >2 classes and crime, while a negative association was found for gender, commuting to school, sleep, household head BMI, place to walk, ease of walking to the bus stop and paved roads ($p<0.1$), and later included in the robust regression analyses. In addition, variables related to MVPA were work, commuting to school, BF, no stress, household head MVPA, frequency of PE >2 classes and crime (all positive), and gender, sleep, place for walking and paved roads (all negative).

The results of the final models of robust linear regression are presented in Table 3. The analysis for LPA resulted in a statistically significant final model [$F_{(12, 283)} = 4.91$; $p<0.001$; $R^2 = 0.2073$]. Individual and environmental variables were able to respectively predict 64% and 13.6% of adolescents' participation in LPA. Work ($\beta_p = 0.2322$; $p < 0.001$) (positive), gender ($\beta_p = -0.1318$; $p = 0.021$), commuting to school ($\beta_p = -0.1501$; $p = 0.008$), sleep ($\beta_p = -0.1260$; $p = 0.017$) and paved roads ($\beta_p = -0.1360$; $p = 0.019$) (all negative) were associated with LPA. Thus, male adolescents, those who work, students who passively commute to school and those who live in unpaved neighborhoods respectively performed 10.2, 23.9, 11.5 and 11.9 minutes more LPA than their respective peers. The results regarding sleep time indicated that there was an average reduction of 5.0 minutes in LPA time for each additional hour of sleep.

The analysis for MVPA resulted in a statistically significant final model [$F_{(11, 284)} = 7.94$; $p<0.001$; $R^2 = 0.2587$]. It was also observed that individual (59.4%) and environmental (27.4%) variables were able to predict adolescents' participation in MVPA. Work ($\beta_p = 0.1656$; $p = 0.007$), commuting to school ($\beta_p = 0.1242$; $p = 0.034$) and crime ($\beta_p = 0.1376$; $p = 0.014$)

**Table 1. Median values and interquartile range of continuous variables and absolute and relative frequency values of categorical variables for the total sample and separated by gender.**

| CONTINUOUS VARIABLES | | | | |
|---|---|---|---|---|
| | **Total** | **Female** | **Male** | **p-value** |
| | Median (P25 –p75) | Median (P25 –P75) | Median (P25 –P75) | |
| Age (years) | 15 (15–16) | 15 (15–16) | 15 (15–16) | 0.445 |
| LPA (min) | 159.5 (129.9–184.4) | 152.4 (126.5–182.1) | 165.6 (134.0–190.1) | 0.034* |
| MVPA (min) | 61.7 (44.8–80.7) | 52.70 (41.5–69.0) | 74.8 (56.5–93.8) | <0.001* |
| ST (h) | 7.0 (4.1–10.6) | 7.7 (4.5–10.8) | 6.42 (3.7–10.1) | 0.158 |
| Sleep (h) | 7.5 (6.9–8.0) | 7.6 (6.9–8.1) | 7.39 (6.9–7.9) | 0.321 |
| Sit Time (h) | 6.4 (4.7–8.6) | 6.6 (4.8–8.9) | 6.33 (4.6–8.6) | 0.316 |
| CATEGORICAL VARIABLES | | | | |
| | **Total** | **Female** | **Male** | **p-value** |
| | **309 (100%)** | **176 (56.9%)** | **133 (43.1%)** | |
| Block 1—Individual | | | | |
| *Work* | | | | 0.003* |
| Does not work[†] | 257 (83.2) | 156 (88.6) | 101 (75.9) | |
| Works | 52 (16.8) | 20 (11.4) | 32 (24.1) | |
| *Commute to school* | | | | <0.001* |
| Passive[†] | 145 (46.9) | 101 (57.4) | 44 (33.1) | |
| Active | 164 (53.1) | 75 (42.6) | 89 (66.9) | |
| *Body fat* | | | | <0.001* |
| Risk of Overweight/Overweight [†] | 143 (46.3) | 114 (64.8) | 29 (21.8) | |
| Low weight/Eutrophic | 166 (53.7) | 62 (35.2) | 104 (78.2) | |
| *Alcohol* | | | | 0.213 |
| Consumes[†] | 93 (30.1) | 48 (27.3) | 45 (33.8) | |
| Does not consume | 216 (69.9) | 128 (72.7) | 88 (66.2) | |
| *Stress* | | | | <0.001* |
| Always[†] | 95 (30.7) | 27 (15.3) | 43 (32.4) | |
| Sometimes | 144 (46.6) | 72 (40.9) | 72 (54.1) | |
| Never | 70 (22.7) | 77 (43.8) | 18 (13.5) | |
| *Loneliness* | | | | <0.001* |
| Always[†] | 63 (20.4) | 11 (6.3) | 33 (24.8) | |
| Sometimes | 202 (65.4) | 118 (67.1) | 84 (63.2) | |
| Never | 44 (14.2) | 47 (26.7) | 16 (12.0) | |
| Block 2—Family | | | | |
| *MVPA_ household head* | | | | 0.013* |
| Inactive[†] | 161 (52.1) | 104 (59.1) | 57 (42.9) | |
| Active | 135 (43.6) | 68 (38.6) | 67 (50.4) | |
| Missing | 13 (4.2) | 4 (2.3) | 9 (6.7) | |
| *BMI_ household head* | | | | 0.792 |
| High[†] | 186 (60.2) | 107 (60.8) | 79 (59.4) | |
| Adequate | 110 (35.6) | 65 (36.9) | 45 (33.9) | |
| Missing | 13 (4.2) | 4 (2.3) | 9 (6.7) | |
| *SS* | | | | 0.069 |
| Low[†] | 60 (19.4) | 27 (15.3) | 33 (24.8) | |
| Medium | 159 (51.5) | 97 (55.1) | 62 (46.7) | |
| High | 77 (24.9) | 48 (27.3) | 29 (21.8) | |
| Missing | 13 (4.2) | 4 (2.3) | 9 (6.7) | |

*(Continued)*

**Table 1.** (Continued)

| | | | | |
|---|---|---|---|---|
| **Block 3—School** | | | | |
| *PE spaces* | | | | 0.601 |
| Only 1[†] | 162 (52.4) | 90 (51.1) | 72 (54.1) | |
| 2 or more | 147 (47.6) | 86 (48.9) | 61 (45.9) | |
| *Extra PA class* | | | | 0.319 |
| No[†] | 118 (38.2) | 63 (35.8) | 55 (41.4) | |
| Yes | 191 (61.8) | 113 (64.2) | 78 (58.6) | |
| *PE class frequency* | | | | <0.001* |
| Excused[†] | 24 (7.8) | 19 (10.8) | 5 (3.8) | |
| Up to 2 classes/week | 263 (85.1) | 152 (86.4) | 111 (83.4) | |
| More than 2 classes/week | 22 (7.1) | 5 (2.8) | 17 (12.8) | |
| **Block 4—Neighborhood** | | | | |
| *Place to walk* | | | | 0.591 |
| No[†] | 63 (20.4) | 34 (19.3) | 29 (21.8) | |
| Yes | 246 (79.6) | 142 (80.7) | 104 (78.2) | |
| *Walk to bus stop* | | | | 0.973 |
| No[†] | 30 (9.7) | 17 (9.7) | 13 (9.8) | |
| Yes | 279 (90.3) | 159 (90.3) | 120 (90.2) | |
| *Paved roads* | | | | 0.114 |
| No[†] | 79 (25.6) | 51 (29.0) | 28 (21.0) | |
| Yes | 230 (74.4) | 125 (71.0) | 105 (79.0) | |
| *Lighting* | | | | 0.029* |
| No[†] | 107 (34.6) | 70 (39.8) | 37 (27.8) | |
| Yes | 202 (65.4) | 106 (60.2) | 96 (72.2) | |
| *Crime* | | | | 0.198 |
| Yes[†] | 129 (41.7) | 79 (44.9) | 50 (37.6) | |
| No | 180 (58.3) | 97 (55.1) | 83 (62.4) | |
| *Commute time* | | | | 0.130 |
| Low (1–5 min) | 204 (66) | 114 (64.8) | 90 (67.7) | |
| Medium (6–10 min) | 67 (21.7) | 45 (25.5) | 22 (16.5) | |
| High (11–30 min)[†] | 22 (7.1) | 10 (5.7) | 12 (9.0) | |
| Missing | 16 (5.2) | 4 (4.0) | 9 (6.8) | |

* Statistical significance;

[†], reference category;

LPA; light physical activity; MVPA, moderate to vigorous physical activity; ST, screentime; Sit Time, sitting time; min, minute; h, hours; BF, body fat; BMI, body mass index; SS, socioeconomic status; PA, physical activity; PE, physical education.

(all positive), and gender ($\beta_p$ = −0.3041; p < 0.001) and paved roads ($\beta_p$ = −0.1357; p = 0.031) (both negative) were associated with MVPA. Thus, male adolescents, those who work, those who actively commute to school, students who live in unpaved neighborhoods and those who have the perception of living in neighborhoods with crime respectively performed 17.6, 12, 7, 7.1, 8.9 and 7.9 more minutes of MVPA than their respective peers.

## Discussion

The main findings of the present study indicated that individual and environment neighborhood predictors were associated with longer time in LPA and MVPA. In addition, some of

**Table 2. Association between individual, family, school and neighborhood variables with LPA minutes per day and MVPA per day using simple linear regression.**

| Variables | LPA | | | | MVPA | | | |
|---|---|---|---|---|---|---|---|---|
| | β | CI | p-value | $R^2$ | β | CI | p-value | $R^2$ |
| **BLOCK 1—INDIVIDUAL** | | | | | | | | |
| Sex—Male[†] | −10.1 | −18.1–−1.0 | 0.021* | 0.017 | −21.4 | −27.4–15.3 | <0.001* | 0.137 |
| Age | 1457.0 | −6.1–9.0 | <0.001 | 0.704 | 4.3 | −1.4–9.9 | 0.137 | 0.007 |
| Work—No[†] | 29.7 | 18.7–40.7 | <0.001* | 0.084 | 20.2 | 11.9–28.5 | <0.001* | 0.070 |
| School commute Passive[†] | −12.6 | −21.1–−4.1 | 0.004* | 0.027 | 8.6 | 2.2–14.9 | 0.008* | 0.023 |
| BF—Risk of overweight/overweight[†] | 5.8 | −2.8–14.4 | 0.185 | 0.006 | 11.3 | 4.9–17.6 | <0.001* | 0.039 |
| Alcohol—No[†] | 8.2 | −1.2–17.5 | 0.086* | 0.010 | 5.7 | −1.3–12.6 | 0110 | 0.008 |
| Always stressed[†] | - | - | - | 0.070 | - | - | - | 0.013 |
| Sometimes stressed | 4.8 | −5.2–14.7 | 0.347 | | 4.7 | −2.7–12.2 | 0.210 | |
| Never stressed | 0.7 | −3.2–20.6 | 0.150 | | 8.9 | 0.1–17.8 | 0.047* | |
| Always lonely[†] | - | - | - | 0.008 | - | 1- | - | 0.011 |
| Sometimes lonely | 0.8 | −10.1–11.6 | 0.893 | | 1.1 | −7.0–9.1 | 0.802 | |
| Never lonely | 10.1 | −4.7–−24.9 | 0.179 | | 9.1 | −1.9–20.2 | 0.104 | |
| Sitting time | −0.3 | −1.7–1.1 | 0.637 | 0.001 | −0.7 | −1.7–0.3 | 0.178 | 0.006 |
| Sleep | −7.5 | −11.8–−3.1 | <0.001* | 0.032 | −3.3 | −6.6–0.1 | 0.053* | 0.012 |
| Screen time | −0.8 | −1.7–−0.2 | 0.102 | 0.009 | −0.51 | −1.22–0.2 | 0.163 | 0.006 |
| **BLOCK 2—FAMILY** | | | | | | | | |
| MVPA_ household head—Inactive[†] | 6.9 | −1.9–15.7 | 0.123 | 0.008 | 9.1 | 2.6–15.6 | 0.006* | 0.025 |
| BMI_ household head—High[†] | −7861.0 | 16.9–1.2 | 0.089* | 0.010 | −2.5 | −9.3–4.3 | 0.465 | 0.002 |
| Low socioeconomic status[†] | - | - | - | 0.004 | - | - | - | 0.013 |
| Medium socioeconomic status | 2.9 | −8.5–14.4 | 0.611 | | −1.9 | −10.4–6.7 | 0.667 | |
| High socioeconomic status | 7.1 | −5.9–20.1 | 0.287 | | 4.9 | −3.7–15.7 | 0.221 | |
| **BLOCK 3—SCHOOL** | | | | | | | | |
| PE spaces—Only 1 space[†] | 1.8 | −6.8–10.4 | 0.680 | 0.001 | −2.5 | −8.9–3.9 | 0.448 | 0.002 |
| Extra PA class—Not offered[†] | 6.8 | −2.0–15.6 | 0.131 | 0.004 | −0.7 | −7.3–5.9 | 0.829 | 0.001 |
| PE Frequency 1[†] | - | - | - | 0.010 | - | - | - | 0.013 |
| PE Frequency 2 | 10.8 | −5.2–26.9 | 0.186 | | 6.8 | −5.2–18.7 | 0.266 | |
| PE Frequency 3 | 19.2 | −2.9–41.4 | 0.090* | | 16.8 | 0.3–33.4 | 0.046* | |
| **BLOCK 4—NEIGHBORHOOD** | | | | | | | | |
| Place to walk—No[†] | −14.9 | −25.7–−4.4 | 0.006* | 0.025 | −8.5 | −16.4–0.6 | 0.035* | 0.014 |
| Walk to bus stop—No[†] | −14.3 | −28.–0.2 | 0.053* | 0.012 | −6.8 | −17.6–4.0 | 0.217 | 0.005 |
| Paved roads—No[†] | −15.6 | −25.3–−5.9 | 0.020* | 0.032 | −6.5 | −13.9–0.8 | 0.080* | 0.010 |
| Lighting—No[†] | −0.9 | −9.9–8.1 | 0.833 | <0.001 | 2.2 | −4.6–8.9 | 0.523 | 0.001 |
| Crime—No[†] | 7.6 | −1.1–16.3 | 0.086* | 0.010 | 7.6 | 1.2–14.1 | 0.020* | 0.001 |
| High commute time to bus stop[†] | - | - | - | <0.001 | - | - | - | 0.004 |
| Medium commute time to bus stop | −9.4 | −27.8–9.0 | 0.316 | | −7.6 | −21.4–6.2 | 0.306 | |
| Low commute time to bus stop | −12.1 | −28.9–4.7 | 0.159 | | −6.6 | −19.2–6.0 | 0.282 | |

\* Statistical significance;

[†] reference category;

β, Beta Non-standard; 95%CI, 95% confidence interval; $R^2$, R squared; BF, body fat; MVPA_household head, moderate to vigorous physical activity of the household head; BMI_household head, body mass index of the household head; PE space, spaces for Physical Education classes; extra-class PA, extra-class of physical activity; PE Frequency 1, attendance of physical education classes_excused; PE Frequency 2, attendance of physical education classes_ up to 2 classes/week; PE Frequency 2, attendance of physical education classes_ more than 2 classes/week.

**Table 3. Robust Linear Regression for the association between individual, family, school, and neighborhood variables with the LPA minutes per day and MVPA minutes per day.**

| | $R^2$ | F | dof1 | dof2 | $\beta$ | $\beta_p$ | t | p-value | 95%CI |
|---|---|---|---|---|---|---|---|---|---|
| **Variables** | | | | | **LPA** | | | | |
| Gender—Male[†] | 0.2073 | 4.91 | 12 | 283 | −10.24 | −0.1318 | −2.33 | 0.021* | −18.89 − −1.59 |
| Work—No[†] | | | | | 23.96 | 0.2322 | 3.91 | <0.001* | 11.89−36.01 |
| Commute to school—Passive[†] | | | | | −11.53 | −0.1501 | −2.67 | 0.008* | −20.02 − −3.04 |
| Alcohol—No[†] | | | | | 2.29 | 0.0274 | 0.54 | 0.589 | −6.05−10.64 |
| Sleep | | | | | −5.00 | −0.1260 | −2.40 | 0.017* | −9.12 − −0.89 |
| BMI_household head—High[†] | | | | | −7.98 | −0.1006 | −1.81 | 0.071 | −16.66−0.69 |
| PE Frequency 3 | | | | | −3.93 | −0.0257 | −0.51 | 0.610 | −19.07−11.22 |
| Place to walk—No[†] | | | | | −9.64 | −0.1011 | −1.67 | 0.096 | −20.99−1.71 |
| Walk—bus stop—No[†] | | | | | −7.48 | −0.0580 | −0.88 | 0.379 | −24.22−9.25 |
| Paved roads—No[†] | | | | | −11.94 | −0.1360 | −2.36 | 0.019* | −21.91 − −1.96 |
| Crime—No[†] | | | | | 5.45 | 0.0699 | 1.29 | 0.197 | −2.84−13.74 |
| constant | | | | | 230.75 | - | 12.02 | <0.001 | 192.96−268.55 |
| **Variable** | | | | | **MVPA** | | | | |
| Gender—Male[†] | 0.2587 | 7.94 | 11 | 284 | −17.63 | −0.3041 | −4.38 | <0.001* | −25.54 − −9.71 |
| Work—No[†] | | | | | 12.74 | 0.1656 | 2.74 | 0.007* | 3.57−21.92 |
| Commute to school—Passive[†] | | | | | 7.12 | 0.1242 | 2.13 | 0.034* | 0.53−13.70 |
| BF_Risk of overweight / Overweight[†] | | | | | 3.16 | 0.0550 | 0.99 | 0.322 | −3.10−9.41 |
| No stress | | | | | −0.06 | −0.0009 | −0.02 | 0.987 | −7.71−7.59 |
| Sleep | | | | | −2.08 | −0.0701 | −1.27 | 0.205 | −5.29−1.14 |
| MVPA_household head—Inactive[†] | | | | | 5.76 | 0.1003 | 1.94 | 0.053 | −0.08−11.60 |
| PE Frequency 3 | | | | | −0.04 | 0.0004 | −0.01 | 0.995 | −13.17−13.08 |
| Place to walk—No[†] | | | | | −5.70 | −0.0801 | −1.48 | 0.141 | −13.31−1.90 |
| Paved roads—No[†] | | | | | −8.88 | −0.1357 | −2.16 | 0.031* | −16.97 − −0.79 |
| Crime—No[†] | | | | | 7.99 | 0.1376 | 2.48 | 0.014* | 1.66−14.33 |
| constant | | | | | 89.88 | - | 6.56 | <0.001 | 62.93−116.83 |

* Statistical significance;

[†], reference category;

$R^2$, R squared; F, F statistic; dof1, degrees of freedom 1; dof2, degree of freedom 2; β, Non-standardized beta; $\beta_p$, standardized Beta; t, t statistic; 95%CI, 95% confidence interval; BF, body fat; ST, Screen Time; MVPA_household head, moderate to vigorous physical activity of the household head; BMI_household head, body mass index of the household head; PE Frequency 3, attendance of physical education classes_more than 2 classes/week.

these predictors were only associated with a certain level of PA. Previous investigations have identified factors that influence PA among adolescents who comply with daily MVPA guidelines [4, 53, 54], to general PA [12, 21] or to active transport/commute to school [55, 56]. The present study advances the literature by presenting individual and environmental predictors that were associated also with LPA in adolescents and that should receive greater attention.

The analyzes found that adolescents had a higher daily MVPA time (average = 65.9 minutes/day) than that recommended by the MVPA guidelines [9, 10]. There are no recommendations for daily LPA values for adolescents [7], which justifies the scarcity of studies that propose to evaluate the factors that can influence LPA performance. Studies of this nature exploring LPA are important, as research has demonstrated the relevance of LPA to adolescent health [8, 57, 58]. The present sample presented an average of 160.2 minutes/day for LPA, constituting lower values than those found by Lopes et al. [59] for Brazilian adolescents, in which the average daily LPA time was 194 minutes, and of these, 55 minutes of LPA were spent

during school time, and 139 minutes of LPA outside school. Regardless of the PA level analyzed, the time in minutes for boys were significantly higher than for girls.

Regarding individual predictors, there were positive associations with the variables of work and commuting to school, and negative associations with the variables of gender, sleep time and for commuting to school. Boys tend to have a greater time spent in LPA and MVPA. This is in line with previous findings that suggest that girls tend to be more inactive than boys [60, 61]. Some studies have found superiority in MVPA [54, 60, 61] and LPA time [59, 60] for boys. There are some justifications for this difference in PA between the genders, such as culturally boys prefer to practice sports and participate in sports competitions, which usually involves vigorous PA, in contrast to girls, who are more inclined to perform activities with little energy expenditure, such as sitting and talking with friends, a trait which can make them less active [62, 63].

Similarly, adolescents who work showed a greater time spent in LPA and MVPA. Systematic reviews on the correlates of PA in general also observed the influence of the work variable on PA of young people [12, 21], mainly related to household chores. It was not possible to investigate what activities were developed during working time by adolescents in the present study and which may have favored the performance of the greatest time in LPA and MVPA. In addition, the studies found that evaluated the correlates of LPA and MVPA [59, 60, 64] did not analyze the association of work with PA in adolescents.

These results are important because they add information to the literature regarding the influence of work on the different PA levels, given the lack of studies of this nature that assess this variable. It is possible that these adolescents who work have to actively commute to work, which contributes to achieving a greater time in LPA. In addition, the activities developed at work may have intensity predictors ranging from mild to moderate-vigorous, which also contributed to the longer time spent in LPA and MVPA in comparison to adolescents who do not work.

The way of commuting to school showed a negative association with LPA and a positive association with MVPA. Adolescents who passively commute to school had more time in LPA, while those who actively commute to school had more time in MVPA. Similar to the findings of this study, active transport was positively associated with MVPA in UK adolescents; however, no association was found for LPA in this same study [64]. It is possible that the adolescents in the present study commute to school at a greater intensity, which contributes to a longer MVPA time throughout the day. In addition, previous research has reported the contribution of active transport to school and/or leisure destinations in increasing overall PA levels in children and adolescents [56, 64–66].

Regarding the negative association with LPA, it may be that adolescents who passively go to school perform other activities throughout their day that contribute to a longer LPA time. In addition, commuting to school only represents one movement form throughout the day in which the adolescent goes to a specific place present in their daily routine, but they can also actively move to other places during the rest of the day, thus contributing to a longer LPA time.

Sleep time was negatively associated with LPA. Thus, there was an average reduction of 5.0 minutes in LPA time for each additional hour of sleep by adolescents. A possible explanation for this result is based on the codependency of movement behaviors (LPA, MVPA, sedentary behavior and sleep) in the finite time of 24 hours. If we consider that the time in a 24h period is finite, the time spent on different movement behaviors are intrinsically collinear and codependent [67, 68]. Thus, any modification in the time spent on a behavior (for example, as occurred with sleep time) only occurs from the modification of the time spent on at least one of the remaining behaviors, as occurred with LPA time [69–71]. Thus, for the adolescents in

the present sample to have healthy health behaviors, it is recommended (based on the health guidelines) that they perform at least a daily average of 60 minutes of MVPA, sleep an average of 8 hours a night, engage in more LPA throughout the day and reduce the time spent in sedentary behavior [9, 10, 72].

Regarding environmental predictors, there was a positive association with the crime variable and negative associations with the variables for walking and paved roads. Adolescents who live in unpaved neighborhoods tend to have a greater time spent in LPA and MVPA. While those who live in neighborhoods with crime performed a greater time in MVPA. Although some studies show that favorable physical environments, such as those with higher residential density with paved and connected streets, with greater security against crime, close to parks and places to walk [12, 28, 73] are associated with longer PA time, this was not observed in the present study.

Such results for the environmental variables were unexpected, and it is difficult to explain these inverse associations in which unfavorable predictors of the neighborhood environment, such as crime, lack of sidewalks and an adequate place to walk were associated with higher PA values for adolescents. We speculate that these neighborhood variables investigated by the NEWS-Y scale, more related to walkability, such as proximity of home to certain places and commercial facilities, paved roads, street lighting, and crime safety, may not interfere very much in adolescents' PA and have a greater influence on adults' PA, since walkability was developed to capture the opportunities to walk in the daily lives of adults [74].

However, it is important to emphasize that Viçosa, the city where the study was carried out, similar to other cities throughout Brazil, is small, in the countryside, does not have structures such as parks, green areas, recreational facilities or squares for doing PA in the neighborhoods capable of stimulating greater interest among adolescents, which makes it difficult to choose the variables of the neighborhood environment to be studied. Perhaps this is a public policy problem in Brazilian cities with these predictors and that deserves greater attention from government officials. Findings from some systematic reviews indicate that the environment in which adolescents live should be designed with adequate PA facilities, such as playgrounds, recreation areas with sports fields and cycling paths, which should be easily accessible and safe to use to promote PA [12, 21, 28].

This study has some limitations that should be mentioned. First, although the number of individuals evaluated met the sample size calculation, the sample size makes it difficult to carry out some association analysis between variables, such as multilevel linear regression analyses. Second, the use of self-reported measures to assess individual, family, school environment and neighborhood predictors, since they depend on the subjects' understanding of the variables being evaluated, although this is widely used in epidemiological studies due to the ease of obtaining information. Finally, although PA has been objectively evaluated (accelerometry), this method does not make it possible to identify the domain in which it is performed.

Strengths of the current study include the use of different PA levels (LPA and MVPA) in a single study. It was not possible to identify any study in the literature which has concomitantly investigated a set of predictors that can influence both PA levels, most previously conducted studies were only dedicated to a single expression of PA (e.g., MVPA, active transport to school or leisure PA). Such scarcity of information on the variables which can influence the different levels of PA intensities can be attributed to the fact that many studies use subjective measures (self-reports) to assess PA due to the ease of application and low cost [75]. In addition, it is important to emphasize the use of the accelerometer for the direct assessment of PA, unlike some studies that use subjective measures. Another important consideration was the inclusion of a set of potential predictors (individual, family, school environment and neighborhood) associated with different PA levels.

## Conclusion

A series of individual predictors and the neighborhood environment were associated with the PA of the evaluated adolescents, and it seems that some predictors are only associated with a certain level of PA. In this study it was verified that the variables of work, gender, commuting to school, sleep and paved roads were associated with LPA and the variables of work, gender, commuting to school, crime and paved roads were associated with MVPA.

In addition, no associations were found between family variables and school environment. Multi-component interventions that involve both schools and the family and neighborhood environment, which incorporate simple changes, such as raising parents' awareness of how they can support their children's PA from an early age so that it lasts for a lifetime can be effective. Further studies are also suggested on the predictors that can influence PA from a socioecological perspective that can contribute to the identification of factors which can help in the development of strategies that promote greater PA practice in its different domains.

## Supporting information

**S1 Table. Spearman's correlation.** V1, Steps; V2, LPA; V3, MVPA; V4, Age; V5, SitT; V6, Sleep; V7, ST; V8, Gender; V9, BF; V10, Work; V11, Commute type; V12, Consumption of alcohol; V13, Stress; V14, Loneliness; V15, Father's education; V16, Father's BMI; V17, Spaces for PE; V18, Extra PA class; V19, Frequency of PE classes; V20, Place to walk; V21, Walk to the bus stop; V22, Paved roads; V23, Lighting; V24, Crime; V25, Commute time to the bus stop. (DOCX)

**S1 File. LPA, light physical activity; MPA, moderate physical activity; VPA, vigorous physical activity; MVPA, moderate to vigorous physical activity; BMI; body mass index; BF, body fat; SS, socioeconomic status.**
(XLSX)

**S1 Checklist.** *PLOS ONE* **clinical studies checklist.**
(DOCX)

## Acknowledgments

The authors would like to thank all the students and by her parents or legal guardians who participated in the study and the teachers, educators and principals who facilitated research to take place.

## Author Contributions

**Conceptualization:** Isabella Toledo Caetano, Fernanda Karina dos Santos, Alynne Christian Ribeiro Andaki, Thayse Natacha Q. F. Gomes, Paulo Roberto dos Santos Amorim.

**Data curation:** Isabella Toledo Caetano, Fernanda Karina dos Santos, Thayse Natacha Q. F. Gomes.

**Formal analysis:** Isabella Toledo Caetano.

**Investigation:** Isabella Toledo Caetano.

**Methodology:** Isabella Toledo Caetano, Fernanda Karina dos Santos, Alynne Christian Ribeiro Andaki, Thayse Natacha Q. F. Gomes, Paulo Roberto dos Santos Amorim.

**Project administration:** Paulo Roberto dos Santos Amorim.

**Supervision:** Fernanda Karina dos Santos, Paulo Roberto dos Santos Amorim.

**Writing – original draft:** Isabella Toledo Caetano.

**Writing – review & editing:** Isabella Toledo Caetano, Fernanda Karina dos Santos, Alynne Christian Ribeiro Andaki, Thayse Natacha Q. F. Gomes, Paulo Roberto dos Santos Amorim.

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
