## [Decision Letter · Decision Letter 0]

11 Jan 2024

PONE-D-23-34450Individual, family, school and neighborhood predictors related to different levels of physical activity in adolescentsPLOS ONE

Dear Dr. Gomes,

Thank you for submitting your manuscript to PLOS ONE. After careful consideration, we feel that it has merit but does not fully meet PLOS ONE’s publication criteria as it currently stands. Therefore, we invite you to submit a revised version of the manuscript that addresses the points raised during the review process.

We look forward to receiving your revised manuscript.

Kind regards,

Hsin-Yen Yen

Academic Editor

PLOS ONE

Journal Requirements:

"This study was financed in part by the Coordination for the Improvement of Higher Education Personnel – Brazil (CAPES) – PNPD – CAPES."

5. We notice that your supplementary [Supplementary Table 1] are included in the manuscript file. Please remove them and upload them with the file type 'Supporting Information'. Please ensure that each Supporting Information file has a legend listed in the manuscript after the references list.

Reviewers' comments:

Reviewer's Responses to Questions

**Comments to the Author**

1. Is the manuscript technically sound, and do the data support the conclusions?

Reviewer #1: Yes

Reviewer #2: Partly

2. Has the statistical analysis been performed appropriately and rigorously? 

Reviewer #1: I Don't Know

Reviewer #2: No

3. Have the authors made all data underlying the findings in their manuscript fully available?

Reviewer #1: Yes

Reviewer #2: No

4. Is the manuscript presented in an intelligible fashion and written in standard English?

Reviewer #1: Yes

Reviewer #2: Yes

5. Review Comments to the Author

Reviewer #1: Dear Authors,

I had a real pleasure in reviewing this very interesting manuscript.

In order to improve it I would reccomend to:

- add some statistcal results in the abstract - expressed in numbers.

-add the clear information on WHO recommendation according to the PA in adolescents in introduction

- add the infomrationon VPA as one of reecoomended by WHO existing mesurements of levels of PA in adolescent with information why yoy havent decided to check the level of VPA in your research.

- add definitions of LPA and MVPA - in the methods section.

Best regards,

Reviewer

Reviewer #2: Dear Authors,

Thank you for the opportunity to review the paper titled Individual, family, school and neighborhood predictors related to different levels of physical activity in adolescents. Here are my comments.

Major Concerns

1. High schools were randomly sampled. If I understand correctly, all students of those high schools were asked to participate. However, because of the sampling of the high schools it is a multistage sample anyway. Because of this sampling method clustering is introduced in the data. Therefore, the school level should have been taken into account in the analyses: multilevel analysis.

2. Perceived school environment predictor are included in the questionnaire, while multiple students per school were asked to participate. These predictors are more similar in students attending the same school than in students attending different schools. This is another reason to include the school level in the analyses. In addition, one might hypothesize that perceived school environment predictor are different in each class; therewith introducing clustering at the class-level, too.

3. On top of that, there is the neighbourhood level. Neighbourhood predictors are also included in the study. I assume that students attending the same school also live in a limited number of neighbourhoods. This introduces clustering per neighbourhood. In addition, depending on the distance between the high schools, students living in one neighbourhood can attend different schools. If so, data do not have a normal multilevel structure, but a cross-level structure. When in the analysis of multilevel data clustering is ignored, standard errors and significances are not correct.

4. Please amend the title to include the research question and study design.

6. PLOS authors have the option to publish the peer review history of their article (what does this mean?). If published, this will include your full peer review and any attached files.

Reviewer #1: No

Reviewer #2: No

---

## [Author Response · Author response to Decision Letter 0]

8 Apr 2024

Viçosa, Minas Gerais, Brazil. April 7th 2024.

Dear Hsin-Yen Yen, Academic Editor, PLOS ONE,

Firstly, we would like to thank the editor and reviewers for their comments/suggestions/contributions to the manuscript entitled “Individual, family, school and neighborhood predictors related to different levels of physical activity in adolescents: a cross-sectional study”. We considered that the comments were extremely important and constructive, contributing to improve the quality of the manuscript. It is important to highlight that, for some of the contributions, we tried to better explain our opinions regarding the subject.

We describe each one of the comments along with our answers, which are highlighted in yellow. The changes in the manuscript are also highlighted in yellow.

 Hoping to meet the quality requirements of PLOS ONE, we are at your disposal for whatever is necessary.

Kind regards.

Journal Requirements:

R. A review was carried out on the manuscript to verify the PLOS ONE style, so modifications were made to the nomenclature referring to the database, which was replaced by “Supporting Information Files”. 

R. Along with the revised manuscript, the minimum set of underlying data of our study called “Supporting Information Files” will be uploaded. Furthermore, the data will be available in the “Public Repository” (https://data.mendeley.com/datasets/gxcs8wzcfb/4).

"This study was financed in part by the Coordination for the Improvement of Higher Education Personnel – Brazil (CAPES) – PNPD – CAPES."

R. The following excerpts were inserted in the cover letter:

 "This study was financed in part by the Coordination for the Improvement of Higher Education Personnel – Brazil (CAPES) – PNPD – CAPES. The funders had no role in study design, data collection and analysis, decision to publish, or preparation of the manuscript." 

R. Along with the revised manuscript, the minimum set of underlying data of our study called “Supporting Information Files” will be uploaded. Furthermore, the data will be available in the “Public Repository” (https://data.mendeley.com/datasets/gxcs8wzcfb/4).

 Furthermore, an excerpt mentioning the supporting information file (Page 12, Lines 294-295) was mentioned in the manuscript.

“The minimum set of underlying data of our study called “Supporting Information Files” was uploaded. The data will be available in the “Public Repository”.”

5. We notice that your supplementary [Supplementary Table 1] are included in the manuscript file. Please remove them and upload them with the file type 'Supporting Information'. Please ensure that each Supporting Information file has a legend listed in the manuscript after the references list.

R. The Supplementary Table has been removed from the manuscript and uploaded in a separate file of the 'Supporting Information' file type.

Additionally, each supporting information file has a caption listed in the manuscript after the list of references (Page 37, Line 803).

 Furthermore, the text references information from the Supplementary Table 1 (Page 13, Line 309).

R. A caption for the “Supporting Information File” has been inserted into the manuscript after the list of references (Page 37, Line 810). 

Furthermore, the text mentions the information file and where it is located (Page 12, Lines 294-295).

“The minimum set of underlying data of our study called “Supporting Information Files” was uploaded. The data will be available in the “Public Repository”.”

Reviewers' comments:

Reviewer #1:

Dear Authors, I had a real pleasure in reviewing this very interesting manuscript.

In order to improve it I would recommend to:

• add some statistical results in the abstract - expressed in numbers.

R. Some statistical results expressed in numbers were inserted in the abstract (Page 2, Lines 42-49).

"Individual and environmental variables were able to respectively predict 64% and 13.6% of adolescents’ participation in LPA. Work (ꞵp = 0.2322), gender (ꞵp = -0.1318), commuting to school (ꞵp = -0.1501), sleep (ꞵp = -0.1260) and paved roads (ꞵp = -0.1360) were associated with LPA. It was also observed that individual (59.4%) and environmental (27.4%) variables were able to predict adolescents’ participation in MVPA. Work (ꞵp = 0.1656), commuting to school (ꞵp = 0.1242) and crime (ꞵp = 0.1376, and gender (ꞵp = - 0.3041) and paved roads (ꞵp = -0.1357 were associated with MVPA.”

• add the clear information on WHO recommendation according to the PA in adolescents in introduction.

R. Information about PA recommendations for adolescents according to the WHO was added to the text (Page 3, Lines 74-77).

“According to the World Health Organization (WHO), adolescents should practice, on average, at least 60 minutes of MVPA per day and incorporate at least 3 days per week of vigorous-intensity aerobic activities, as well as those that strengthen muscles and bones [10]”.

• add the information on VPA as one of recommended by WHO existing measurements of levels of PA in adolescent with information why you haven’t decided to check the level of VPA in your research.

R. Please, note that according to WHO guidelines, adolescents are advised to engage in a minimum average of 60min/day of moderate-to-vigorous PA, rather than a minimum requirement specifically for VPA (with an emphasis on a combination of both moderate and vigorous PA). With this in mind, our focus was on investigating MVPA, and not MPA and VPA separately.

In addition, the WHO guidelines advocate for the principle that “every movement counts”, encouraging, when possible, the reduction of sedentary behaviour, by substituting it with LPA. Taking this into account, in addition with the fact that available evidence supports the health benefits of LPA (and not only MVPA), we also included LPA in the analysis.

In summary, our decision to investigate MVPA and LPA aligns with WHO guidelines and current scientific literature/evidence. We hope the reviewer understands our decision.

Information regarding the VPA recommendations was added to the text when we responded to the previous question, regarding the WHO recommendations on PA for adolescents (Page 3, Lines 74-77).

“According to the World Health Organization (WHO), adolescents should practice, on average, at least 60 minutes of MVPA per day and incorporate at least 3 days per week of vigorous-intensity aerobic activities, as well as those that strengthen muscles and bones [10]”.

• add definitions of LPA and MVPA - in the methods section.

R. The definitions of LPA and MVPA have been inserted into the text (Page 6; Lines 139-141).

“ActiGraph GT3X accelerometer was used to monitor the time spent in light PA (LPA) and moderate-to-vigorous PA (MVPA) (min.day-1), and the sleep/wake time (hours.day-1).”

Reviewer

Reviewer #2: Dear Authors, thank you for the opportunity to review the paper titled Individual, family, school and neighborhood predictors related to different levels of physical activity in adolescents. Here are my comments.

Major Concerns 

1. High schools were randomly sampled. If I understand correctly, all students of those high schools were asked to participate. However, because of the sampling of the high schools it is a multistage sample anyway. Because of this sampling method clustering is introduced in the data. Therefore, the school level should have been taken into account in the analyses: multilevel analysis.

R. There are some assumptions that must be taken into account for the adoption of multilevel analysis: the random selection of components at each level; a minimum number of clusters at level 2 and above; and a minimum number of subjects within each cluster (Austin, 2010; Bell et al., 2010; Austin e Leckie, 2018). 

The random selection of components at each level aims to prevent selection bias in the results. In our study, subjects (level 1) were randomly selected, as described in the methods section. However, the same procedure was not applied during the school selection process, i.e., they were not chosen randomly, but rather because they were the only ones in the city offering public secondary education. And based on this selection process, including these schools in the second level of multilevel analysis would violate one of the assumptions of the analysis.

Furthermore, regarding the number of observations within each level, while different authors propose varying numbers of observations (Austin 2010; Bell et al., 2010; Austin e Leckie, 2018), it is generally recommended that this number must be, at least, greater than 10 (Austin, 2010), (some authors suggest at least 20 (Austin e Leckie, 2018), or even 30 (Bell et al., 2010)). This is because it is the minimum required for accurately estimating the variance component at each level. In our study, we only sampled six schools, which would not have provided a sufficient number of clusters for conducting the multilevel analysis.

However, we were cognizant of the role of the school on the results, and with this in mind we performed the analysis adjusting for the cluster (i.e., school), as has been done in previous studies where the multilevel structure of the data was presented, but the multilevel analysis could not be conducted due to failure to meet the required assumptions. 

2. Perceived school environment predictor are included in the questionnaire, while multiple students per school were asked to participate. These predictors are more similar in students attending the same school than in students attending different schools. This is another reason to include the school level in the analyses. In addition, one might hypothesize that perceived school environment predictor are different in each class; therewith introducing clustering at the class-level, too.

R. We have previously outlined the reasons for not using the multilevel analysis, considering school as a level. Since we were unable to conduct the analysis considering school as the second-level, it makes not possible to include the class as a third-level.

Furthermore, within each school, students were sampled from the same class, which means the absence of within-school differences regarding class.

3. On top of that, there is the neighborhood level. Neighborhood predictors are also included in the study. I assume that students attending the same school also live in a limited number of neighborhoods. This introduces clustering per neighborhood. In addition, depending on the distance between the high schools, students living in one neighborhood can attend different schools. If so, data do not have a normal multilevel structure, but a cross-level structure. When in the analysis of multilevel data clustering is ignored, standard errors and significances are not correct.

R. The sample comes from 39 different neighbourhoods. However, in 51% of these neighbourhoods there were five or fewer adolescents residing. According to Austin (2010), when there are five or fewer observations per cluster, the estimates of the variance (both, dependent variable and standard error) are prone to be inefficient.

Once again, considering the assumptions of the multilevel analysis (Austin, 2010; Bell et al., 2010; Austin e Leckie, 2018), our data does not meet the requirements (minimum number of clusters, and minimum number of observations in each cluster), which would lead to an inadequate estimates of the variance components, standard errors, confidence intervals, and the overall model. Therefore, the adoption of the Ordinary Least Squares (OLS) estimates was our choice, as it allowed us to estimate the coefficients of the final model, with the adjustment for clustering, without diminishing the explanatory power of the model.

4. Please amend the title to include the research question and study design.

R. The title was changed to include the study design (Page 1; Lines 4-5).

“Individual, family, school and neighborhood predictors related to different levels of physical activity in adolescents: a cross-sectional study.”

References

Austin P. C., Leckie G. The effect of number of clusters and cluster size on statistical power and Type I error rates when testing random effects variance components in multilevel linear and logistic regression models. Journal of Statistical Computation and Simulation, 2018, 88:16, 3151-3163. DOI: 10.1080/00949655.2018.1504945.

Austin PC. Estimating multilevel logistic regression models when the number of clusters is low: a comparison of different statistical software procedures. Int J Biostat. 2010, 22;6(1). DOI: 10.2202/1557-4679.1195.

Bell B. A, Morgan G. B., Kromrey J. D., Ferron J. M. The Impact of Small Cluster Size on Multilevel Models: A Monte Carlo Examination of Two-Level Models with Binary and Continuous Predictors. Section on Survey Research Methods – JSM 2010. 

http:/

---

## [Decision Letter · Decision Letter 1]

17 May 2024

Individual, family, school and neighborhood predictors related to different levels of physical activity in adolescents: a cross-sectional study

PONE-D-23-34450R1

Dear Dr. Gomes,

We’re pleased to inform you that your manuscript has been judged scientifically suitable for publication and will be formally accepted for publication once it meets all outstanding technical requirements.

Kind regards,

Hsin-Yen Yen

Academic Editor

PLOS ONE

Additional Editor Comments (optional):

Reviewers' comments:

Reviewer's Responses to Questions

**Comments to the Author**

1. If the authors have adequately addressed your comments raised in a previous round of review and you feel that this manuscript is now acceptable for publication, you may indicate that here to bypass the “Comments to the Author” section, enter your conflict of interest statement in the “Confidential to Editor” section, and submit your "Accept" recommendation.

Reviewer #1: All comments have been addressed

2. Is the manuscript technically sound, and do the data support the conclusions?

Reviewer #1: Yes

3. Has the statistical analysis been performed appropriately and rigorously? 

Reviewer #1: Yes

4. Have the authors made all data underlying the findings in their manuscript fully available?

Reviewer #1: Yes

5. Is the manuscript presented in an intelligible fashion and written in standard English?

Reviewer #1: Yes

6. Review Comments to the Author

Reviewer #1: Thank you for implementing my comments I hope you have found them as improving the manuscript.

Best regards

7. PLOS authors have the option to publish the peer review history of their article (what does this mean?). If published, this will include your full peer review and any attached files.

Reviewer #1: No

---

## [Editor Report · Acceptance letter]

24 May 2024

PONE-D-23-34450R1 

PLOS ONE

Dear Dr. Gomes, 

I'm pleased to inform you that your manuscript has been deemed suitable for publication in PLOS ONE. Congratulations! Your manuscript is now being handed over to our production team.

Kind regards, 

on behalf of

Dr. Hsin-Yen Yen 

Academic Editor

PLOS ONE